# Cytoskeletal Organization Correlates to Motility and Invasiveness of Malignant Mesothelioma Cells

**DOI:** 10.3390/cancers13040685

**Published:** 2021-02-08

**Authors:** Maureen Keller, Katarina Reis, Anders Hjerpe, Katalin Dobra, Pontus Aspenström

**Affiliations:** 1Rudbeck Laboratory, Department of Immunology, Genetics and Pathology, Uppsala University, SE-751 85 Uppsala, Sweden; maureen.keller@igp.uu.se; 2Department of Microbiology, Tumor and Cell Biology, Karolinska Institutet, SE-171 77 Stockholm, Sweden; kinareis1969@gmail.com; 3Department of Laboratory Medicine, Division of Pathology, Karolinska Institutet, Huddinge, SE-141 57 Stockholm, Sweden; anders.hjerpe@sll.se (A.H.); katalin.dobra@ki.se (K.D.)

**Keywords:** malignant mesothelioma, cytoskeleton, actin dynamics, vimentin, YAP

## Abstract

**Simple Summary:**

The cytoskeleton is responsible for maintaining normal tissue homeostasis by a tight regulation of cell morphogenesis and cell migration. This homeostasis is lost in cancer mainly because alterations in cytoskeletal dynamics are leading to an increased migratory and invasive capacity of cancer cells. The organization of the cytoskeleton is by large an unknown factor in malignant mesothelioma; therefore we sought to examine the cytoskeletal dynamics and invasive properties of different malignant mesothelioma cell lines originating from patients. Our data suggest that it is possible to classify malignant mesothelioma cell lines into separate categories using straight forward cell staining and analysis of the morphological and invasive capacity of mesothelioma cells. Early diagnosis and new diagnostic tools are urgently needed to effectively treat patients and we propose that the analyses described in this article could potentially provide diagnostic tools that can be further tested on patients.

**Abstract:**

Malignant mesothelioma (MM) is a rare but highly aggressive cancer that primarily originates from the pleura, peritoneum or pericardium. There is a well-established link between asbestos exposure and progression of MM. Direct invasion of the surrounding tissues is the main feature of MM, which is dependent on dysregulated communication between the mesothelium and the microenvironment. This communication is dependent on the dynamic organization of the cytoskeleton. We have analyzed the organization and function of key cytoskeletal components in MM cell lines of increasing malignancies measured as migratory and invasive properties, and we show that highly malignant and invasive MM cells have an organization of the actin filament and vimentin systems that is distinct from the less malignant MM cell lines. In addition, the Hippo tumor suppressor pathway was inactivated in the invasive MM cells, which was seen as increased YAP nuclear localization.

## 1. Introduction

Malignant mesothelioma (MM) is a tumor that is derived from the mesothelial cells that cover all of the body cavities. MMs most often develop in the pleura, peritoneum, and pericardium [1]. Pleural MM is associated with exposure to asbestos or other mineral fibers, whereas some abdominal MMs have a different biology and a more indolent disease course [2]. In the US, 19,011 cases of mesothelioma were diagnosed during 2003–2008, i.e., around 3200 cases per year. This means an annual incidence rate of 1.05 cases per 100,000 population [3]. Characteristically, MM tumors have extremely aggressive local growth that efficiently invades the surrounding tissue and vital organs before it gives rise to metastasis. There are three phenotypic variants of MM: epithelioid, sarcomatoid, and biphasic; where the epithelioid and sarcomatoid components are admixed in the same tumor. Although, all three phenotypes are highly malignant, the epithelioid phenotype indicates better prognosis, whereas the fibroblast-like sarcomatoid phenotypes are considered to be a hallmark of poor prognosis [1]. Changes in the behavior of these tumor cells have predominantly been linked to altered genetic signatures, which is commonly determined by transcriptome analysis using gene-array studies or RNA sequencing [4]. Early diagnosis of MM and new diagnostic tools are urgently needed to effectively treat patients with MM.

The morphological diagnosis of MM is difficult as it can mimic many other conditions, which can be both malignant and benign. Homozygous deletion of the *p16* gene and tumor invasion into fatty tissue are considered the most reliable features of a malignant tumor. Based on their morphology, well-differentiated MMs are difficult to distinguish from reactive mesothelial proliferations and from epithelial tumors that have metastasized to the pleura, which are most often adenocarcinomas. Immunophenotyping effectively separates tumors of mesothelial origin from true epithelial tumors, as they differ in their molecular signatures and expression of cell surface markers. Thus, the presence of mesothelin, calretinin, and podoplanin (recognized by the D2-40 monoclonal antibody), and positive nuclear staining of WT1 indicate a tumor of mesothelial lineage. Tumors that express BerEp4, MOC31, CEA, TTF-1 and neuroendocrine markers represent metastatic tumors of true epithelial origin, whereas lack of nuclear BAP1 is considered a reliable molecular evidence to identify malignant pleural MM. Diagnostic guidelines recommend a combination of at least four markers; two in favor of and two excluding the possibility of MM [5,6].

Despite the advent of molecular methods, tumor morphology remains the best indicator of an aggressive disease course, as the presence of sarcomatoid differentiation worsens the prognosis. Although less frequent, sarcomatoid differentiation constitutes a clear diagnostic and therapeutic challenge, as this subtype is resistant to chemotherapy [2]. Its presence is also a contraindication for surgical resection, as it is difficult to obtain complete surgical removal. Attempts to stratify these tumors further according to their nuclear grade and the absence or presence of molecular markers might allow better adaptation of therapeutic approaches to the molecular features of each individual tumor [7]. Functional markers that predict individual tumor aggressiveness are largely lacking, despite extensive knowledge about the locomotory machinery of tumors in general. Cell lines and ex-vivo systems established from pleural effusions can be used both for extensive molecular characterization and to predict sensitivity and resistance profiles to a wide range of chemotherapeutic agents [8,9,10,11,12].

Studies on triple-negative breast cancer have previously indicated significant differences in cytoskeletal organization between healthy cells and highly invasive cells [13,14]. Here, we aimed to link the motile properties of MM cell lines and rearrangements of the cytoskeletal components, to be able to predict more or less aggressive tumor growth. Seven malignant mesothelioma cell lines and one mesothelial cell line were tested for their cytoskeletal features and their migratory and invasive properties. Interestingly, the cytoskeletal organization and migratory properties differed significantly between these cell lines.

## 2. Results

### 2.1. Organization of Actin Filaments and Focal Adhesions in the MM Cell Lines

To analyze the organization of the actin filament system, MM cells were seeded on coverslips and cultured for 24 h. After this time, the cells were fixed and stained with tetramethyl rhodamine isothiocyanate (TRITC)-conjugated phalloidin to visualize filamentous actin (F-actin). The cells were examined by fluorescence microscopy for the formation of stress fibers, actin arcs, broad lamellipodia and small lamellipodia. The cell lines have previously been characterized as epithelioid (MeT-5a and Mero-25), sarcomatoid (DM-3) or biphasic (M-14-K, JL-1, STAV-AB, STAV-FCS, ZL34) [12,15,16]. MeT5a is considered to be non-malignant but it should be noted that it is immortalized by transfection of the SV40 large T antigen [15]. MeT-5a and Mero-25 have clear epithelial characteristics, they grow in colonies and form a border of cortical actin arcs at the cell periphery (Figure 1A). They are also quite small and round with aspect ratios around 1.6. In contrast, DM-3 very clearly exhibit sarcomatoid, fibroblast-like characteristics: the cells are long (aspect ratio of around 3.6) with long and prominent stress fibers extending the length of the cells (Figure 1A,C,D). The other cell lines are all characterized as predominantly biphasic but the analysis of the F-actin organization reveal that they are very different from each other. JL-1 and STAV-FCS have prominent stress fibers and they are significantly more elongated than the epithelioid cells with aspects ratios of 3.0 for STAV-FCS and 2.6 for JL-1 (Figure 1D). STAV-AB cells differ substantially from all other cell lines since they have broad lamellipodia at the cell edges, M-14-K grow in colonies, like the epithelioid cell lines and ZL34 have short stress fibers and small and condensed membrane ruffles or lamellipodia (Figure 1A). Another feature that could distinguish epithelioid cells from fibroblast-like cells is the presence of a perinuclear actin cap, which is a structure consisting of short actin filaments positioned on top of the cell nucleus [17]. Cancer cells have been found to lack actin caps. Human BJ fibroblasts, included here as a control, had well-developed perinuclear actin caps, whereas this structure was absent from MeT-5a, Mero-25, STAV-AB and ZL34 cells (Appendix A). In DM-3, M-14-K and STAV-FCS cells, perinuclear actin caps were seen in many, but not all cells, whereas JL-1 cells had well-developed perinuclear actin caps, just like in fibroblasts (Appendix A).

We next analyzed the substrate attachment points of the MM cells. To visualize focal adhesions, the cells were stained with an antibody against phosphotyrosine (PY99), which produced a distinct staining of the cell:substrate contact points. The staining revealed that most MM cell lines had focal adhesions at the ends of stress fibers and at the cell periphery but the antibody also detected immature attachment points, so-called focal contacts (Figure 2A). The size of the focal adhesions was measured from microscopy images employing the ImageJ software. Five of the cell lines had focal adhesions of roughly the same size: Met-5a (1.20 μm^2^), Mero-25 (1.15 μm^2^), M-14-K (1.18 μm^2^), STAV-FCS (1.18 μm^2^) and DM-3 (1.11 μm^2^) (Figure 2B). Three cell lines differed significantly: JL-1 had larger focal adhesions of 1.49 μm^2^ and STAV-AB and ZL34 had smaller focal adhesions of 0.97 μm^2^ and 0.91 μm^2^, respectively (Figure 2B). This is in agreement with these cell types having less stress fibers.

### 2.2. Organization of Microtubules and Stable Microtubules in the MM Cell Lines

To visualize organization of the microtubules (MTs), the MM cells were stained with an antibody against α-tubulin. There were marked differences in the organization of microtubules between the different cell lines, MeT-5a, M-14-K, JL-1, STAV-FCS and DM-3 had prominent MTs all the way to the cell periphery (Figure 3). Mero-25 cells also had an accumulation of MTs at the cell edge, but they did not reach all the way to the cell edge due to the presence of the broad zone of cortical actin arcs (Figure 3). STAV-AB and ZL34 differed in the MT organization. In both of these cell lines, only a few individual MTs reached to the cell edge, which could indicate that their MTs are under a more dynamic regime.

The MTs are under a scheme of constant reconstruction, individual MTs undergo cycles of growth and shrinkage in a process known as dynamic instability [18]. More long-lived stable MTs can be modified by acetylation and the presence of acetylated MTs is a hallmark for stable, i.e., less dynamic, MTs [19]. We analyzed MT acetylation with an antibody against acetylated α-tubulin and determined the fraction of acetylated MTs by microscopy analysis (Figure 4A). Three cell lines had a high proportion of acetylated MTs: 68 ± 8% (Mero-25), 72 ± 7% (JL-1) and 78 ± 10% (DM-3), and two cell lines had significantly lower amount of acetylated MTs: 31 ± 10% (M-14-K) and 34 ± 7% (ZL34) (Figure 4B). The other cell lines did not significantly differ from the normal mesothelial MeT-5a cell line: 48 ± 10% (MeT-5a), 55 ± 9% (STAV-AB) and 45 ± 9% (STAV-FCS) (Figure 4B). We also visualized MT ends with an antibody against the end-binding protein EB3. DM-3, STAV-FCS and JL-1 had well-spread MT ends, whereas M-14-K, STAV-AB and ZL34 had fewer and smaller EB3 positive speckles (Appendix A) In the normal Met-5a cell line, the EB3 clusters had the appearance of arrowheads, similar to focal adhesions. Mero-25 cells did not seem to express EB3 at detectable levels (Appendix A). Taken together, this shows that the microtubules in M-14-K and ZL34 are more prone to be under dynamic instability, which could indicate an increased migratory state.

### 2.3. Organization of Vimentin Intermediate Filaments in the MM Cell Lines

We next determined whether the organization of the intermediate filament system differed between the MM cell lines to the same extent as the microfilament system. The cells were stained with antibodies against vimentin, which is a major intermediate filament protein [20]. Vimentin turned out to be expressed in all eight cell lines and the portion of the cell occupied by vimentin filaments was determined by microscopy analysis (Figure 5A,B). In the normal mesothelial MeT-5a cell line, vimentin was predominantly localized in the perinuclear area and occupied 23 ± 5% of the cell body. A similar analysis revealed that 30 ± 7% in Mero-25, 25 ± 5% in JL-1 and 23 ± 4% in STAV-AB of the cells were occupied by vimentin filament, which did not differ significantly from the MeT-5a cell line (Figure 5B). In contrast, vimentin filaments occupied a significant bigger area in ZL34, 48 ± 6% and in DM-3, 63 ± 12%, whereas the numbers in M-14-K and STAV-FCS were somewhere in between the outliers: 37 ± 4% and 38 ± 10%, respectively (Figure 5B). We also stained cells with an antibody against the intermediate filament protein nestin. We could detect nestin expression in the STAV-AB and DM-3 cell lines, but nestin formed filaments only in the sarcomatoid cell line DM-3 (Appendix A).

### 2.4. Organization of Cell:Cell Contacts in the MM Cell Lines

To visualize cell:cell contacts, the MM cell lines were stained with an antibody against β-catenin. This protein links to cadherins and to α-catenin at adherence junctions, which is the core protein complex in these cell:cell contact areas [21]. In the MeT-5a, Mero-25, M-14-K and STAV-AB cell lines, β-catenin was accumulated at the cell junctions, which is the predominant β-catenin localization found in epithelial cells (Figure 6). These findings indicate that these cell lines have epithelial cell-like properties. In contrast, the JL-1, STAV-FCS, ZL34 and DM-3 had fewer points of contacts and β-catenin was found at the cell periphery and in membrane ruffles and to a lesser extent in cell junctions (Figure 6). This mode of β-catenin organization is usually associated with fibroblastic cells [21].

### 2.5. YAP Nuclear Localization in the MM Cell Lines

Yes-associated protein 1 (referred to as YAP throughout the text) is a transcriptional co-activator together with the transcriptional enhancer associate domain (TEAD) transcription factor [22]. YAP is a component in the Hippo tumor suppressor pathway that among other things is involved in the regulation of cell proliferation and migration. When the Hippo pathway is active, i.e., under non-proliferative conditions, YAP is trapped in the cytoplasm and degraded through a ubiquitin-dependent protein-degradation process. When the Hippo pathway is inactivated, for instance during tumor progression, YAP enters the nucleus and can trigger the cell-proliferation machinery [22]. To analyze the status of YAP, we monitored YAP nuclear localization by immunofluorescence microscopy using an antibody against YAP and quantified the proportion of the nuclear YAP of total YAP (Figure 7A). Four cell lines had equal amount of nuclear YAP: (38 ± 3% in Met-5a, 40 ± 4% in Mero-25, 40 ± 6% in M-14-K and 38 ± 13% in STAV-AB (Figure 7B). Three cell lines had significantly less nuclear YAP: 18 ± 9% in JL-1, 21 ± 6% in STAV-FCS and 12 ± 5% in DM-3, whereas ZL34 had significantly more nuclear YAP, 50 ± 6%. The nuclei of the ZL34 cells differed markedly from the other cell lines, they were peanut-shaped and had often multiple lobes (Figure 7A). This difference can be seen as a decreased circularity compared to the other cell lines, with the exception of DM-3 (Figure 7C). In addition, the nuclear perimeter and aspect ratio were higher in ZL34 cells (Figure 7D,E). The nuclei in DM-3 cells also had a higher nuclear perimeter, but the nuclei were generally much bigger than in the other MM cell lines, which most probably is a result of the fact that DM-3 cells are significantly larger than the other MM cell lines (Figure 1B, Appendix A).

### 2.6. Migratory and Invasive Properties of the MM Cell Lines

Next, the migratory and invasive properties of the MM cell lines were tested. First, we investigated how the MM cells behaved in the so-called wound-closure assay. In this case, the wound-closure was analyzed using the IncuCyte imaging system. Cells were seeded in 96 well-plates, and the Wound Maker device was used to inflict wounds of equal size, after which the wound closure was monitored for 48 h. The DM-3 cells could not be studied by this method since the cells did not form confluent monolayers. The other cell lines could be divided into three groups: the normal mesothelial cell line MeT-5a reached 80% wound density after 44 h and wound of STAV-FCS cells were only closed to 66% after 48 h (Figure 8A). In contrast, STAV-AB and ZL34 reached 80% wound density after 15.5 h and 18.5 h, respectively. The remaining three cell lines tested had an intermediate efficiency to reach 80% relative wound density: M-14-K 33 h, JL-1 29.5 h and Mero-25 26.5 h (Figure 8A).

We next tested the ability of the MM cell lines to degrade extracellular matrix proteins. To study this, cells were seeded on coverslips coated with FITC-labeled gelatin. The cells were incubated for 48 h, fixed and the coverslips were analyzed for the matrix degrading activity, visible as the occurrence of areas on the coverslips lacking FITC-gelatin. Both JL-1 and DM-3 cells demonstrated efficient matrix degrading activities and STAV-AB cells also produced black spots indicative of matrix degradation (Figure 8B). However, all other MM cell lines tested had no observable matrix degrading activity (Figure 8B).

Finally, the in-vitro invasion of the MM cell lines was tested. To that end, the cells were analyzed for invasion in a Boyden type of chamber coated with Matrigel. The cells were seeded on top of the Matrigel in the upper chamber, and their migration through the extracellular matrix proteins in the Matrigel was determined by counting the cells that had passed through the Matrigel coat. The ZL34 cells (175 ± 15) were particularly efficient in passing through the Matrigel compared to M-14-K (48 ± 19), STAV-AB (33 ± 12) and Met-5a (30 ± 23) (Figure 8C). The other MM cell lines had a low invasive capacity: Mero-25 (15 ± 4), STAV-FCS (7 ± 4). Interestingly, JL-1 and DM-3 cells, which had a high matrix degradation activity, were more or less unable to migrate through the Matrigel, (7 ± 6) and (4 ± 3), respectively (Figure 8C).

## 3. Discussion

We noted significant differences in the organization of actin filaments between the eight different cell lines studied. The analysis, showed in Figure 1E, showed that MeT-5a and Mero-25 were clearly epithelioid with marked cortical actin arcs and that DM-3 had a clear sarcomatoid phenotype with prominent stress fibers. Both STAV-AB and M-14-K have previously been considered to be of the epithelioid type, but their actin organization is markedly different from MeT-5a and Mero-25 cell lines. M-14-K grow in colonies, like epithelial cells, but are more elongated and have more stress fibers than MeT-5a and Mero-25. The presence of broad lamellipodia in STAV-AB cells does not agree with epithelioid characteristics and the cells grow predominantly as individual cells and not in colonies. The three cell lines characterized as biphasic do not represent a homogenous group, both STAV-FCS and JL-1 have clear sarcomatoid features, such as an abundancy of stress fibers and the presence of actin caps. ZL34 represents its own unique cell type with numerous filopodia and condensed lamellipodia. In addition, ZL34 contains an abundance of actin dots that are highly dynamic when analyzed by live cell imaging (data not shown). Thus, analysis of actin organization and dynamics shows that the biphasic MM cells differ quite considerably from each other, which can provide information about how to treat the disease.

Vimentin has been shown to be a common intermediate filament protein in all MM cell lines [23]. All MM cell lines tested in the present study expressed vimentin, and differences in the organization are likely to have impact on the function of the cells. In MeT-5a, Mero-25, M-14-K, JL-1 STAV-AB and STAV-AB, the vimentin filaments were wrapped around the perinuclear area. In contrast, in ZL34 cells, individual vimentin filaments stretched out to the cell periphery in an aster-like fashion and in DM-3, vimentin filaments often reached the peripheral parts of the cells. Vimentin is an important factor for the regulation of cell stiffness [24]. Highly metastatic breast cancer cells have been found to be markedly softer than non-transformed breast epithelial cells [14]. The vimentin organization in ZL34 and DM-3 cells might indicate that they are less stiff than the other MM cell lines tested in this study. Analysis of cell:cell contacts can provide useful information regarding whether the cells should be classified as epithelioid or not. This way, Met-5a, Mero-25, M-14-K and, to some extent, also STAV-AB, had epithelial cell-like characteristics. The ZL34 cells had very few cell:cell contacts, as determined by staining of the cells with β-catenin. β-catenin has also been shown to translocate to the nucleus upon cell transformation; however, nuclear β-catenin could not be detected for any of the MM cell lines [21].

The microtubule organization differed between the MM cell lines. We detected essentially two different types of organizations: In Mero-25, M-14-K, JL-1, STAV-FCS and DM-3 cells, the microtubules were abundant all the way to the cell periphery. In MeT-5a, STAV-AB and ZL34 the microtubules accumulated at the perinuclear area and much fewer microtubules could be seen at the cell periphery. The most likely interpretation was that there were differences in the dynamic instability between the MM cell lines. Indeed, significant differences in the amount of acetylated microtubules, indicative of stable microtubules [19], could be seen. Mero-25, JL-1 and DM-3 had significantly more acetylated microtubules compared to the normal MeT-5a cell line, suggesting that they are less dynamic. In contrast, M-14-K and ZL34 had less acetylated microtubules indicative of a higher dynamic capacity, which is in line with ZL34 being highly efficient in wound closure and in invasion.

The Hippo pathway serves as a key tumor suppressor, and under normal conditions and when the pathway is activated, the effector protein YAP is sequestered in the cytoplasm predominantly in its phosphorylated state [22]. In cancers, the Hippo pathways is inactive, and YAP can translocate to the nucleus and become transcriptionally active. Several lines of evidence have indicated roles for the Hippo pathway in MM, and a common mechanism appears to be linked to inactivation of the *NF2* gene, which encodes a protein called Merlin [25,26]. Merlin is a cytoskeletal-interacting protein that is related to the Ezrin-Radixin-Moesin (ERM) family of proteins. In contact-inhibited cells, Merlin serves as a component in a polarity complex, and supports the inactivated state of the Hippo pathway [27]. However, *NF2* is frequently inactivated in MM, and as a result, the amount of nuclear YAP increases. In the present study, analysis of YAP nuclear localization was a useful indicator of the aggressiveness of the different MM cell lines. In the ZL34 cells, which were highly invasive and motile, the majority of YAP was in the nucleus. Mero-25, M-14-K and STAV-AB had not significantly more nuclear YAP than the non-tumorigenic MeT-5a cell line. In contrast, significantly less nuclear YAP was detected in the JL-1, STAV-FCS and DM-3 cell lines and they predominantly had a cytoplasmic localization of YAP. Thus, the subcellular localization is an important indicator that can provide information regarding the status of MM cells.

One important question is, does the differences in microtubule stability reflect the migratory capacities of the MM cell line. Yes, there is a clear correlation for some of the cell lines: ZL34 efficiently migrated in the wound closure assay and in the Matrigel invasion assay, whereas DM-3 had a very low invasive capacity. In contrast, DM-3 and JL-1 had a very high matrix degrading activity in the FITC-gelatin degradation assay but the ZL34 cells had no detectable matrix degrading capacity. STAV-AB were somewhere in between these extremes since it efficiently closed the wounds but also had a low, but detachable matrix degrading capacity. Tumor-cell invasion is a well-orchestrated multistep process that drives cancer progression and metastasis. Tumor cells can migrate either as individual cells or as a collective cell mass. Furthermore, individually migrating tumor cells can use two essentially distinct modes of migration: mesenchymal and amoeboid cell migration [28]. The mesenchymal mode of motility is characterized by well-developed focal adhesions and activity of the matrix metalloproteinases (MMPs), which facilitates degradation of the extracellular matrix. In contrast, cells using amoeboid migration move independently of integrin signaling and focal adhesions, and do not require activity of the MMP to invade and metastasize. Many tumor-cell types can switch between these mesenchymal and amoeboid modes of cell motility [29]. The differences in matrix degrading capacity and 2D and 3D migration found between the MM cell lines could reflect the different modes of migration/invasion. DM-3 and JL-1 cells that have a high matrix degrading activity but low invasive capacity could still spread once the extracellular matrix is degraded. These cells represent a sarcomatoid mode of spreading. ZL34 cells, on the other hand, invade efficiently in the absence of any matrix degrading activity. STAV-AB cells are also highly motile and exhibit matrix-degrading activity and can invade into extracellular matrix with intermediate speed. ZL34, and to some extent STAV-AB cells, thus represents the amoeboid type of migration. We noted that the ZL34 cells had increased numbers of filopodia-like protrusions and small and condensed membrane ruffles. It has been shown that cells use filopodia during migration in a three-dimensional environment; for instance, when malignant cancer cells escape from the primary tumor [30,31]. Malignant pleural mesothelioma cells have previously been shown to have long microvilli, which indicates that the presence of filopodia-like protrusions might be a common feature of invasive MM cells [32].

## 4. Materials and Methods

### 4.1. Antibodies and Constructs

The following commercial antibodies and reagents were used: mouse monoclonal anti–α-tubulin (T9026); mouse monoclonal anti-vimentin (V6630); mouse monoclonal anti-acetylated α-tubulin (T6793); tetramethyl rhodamine isothiocyanate (TRITC)-conjugated phalloidin (P1951); 4′,6-diamidino-2-phenylindole dihydrochloride (DAPI, D9542) (Sigma-Aldrich, St. Louis, MO, USA); mouse monoclonal anti–β-catenin (#610153; BD Biosciences, Franklin Lakes, NJ, USA); mouse monoclonal anti-YAP (SC-101199) and mouse monoclonal anti-phosphotyrosine (PY99, SC-7020) (Santa Cruz, Santa Cruz, CA, USA); rabbit polyclonal anti–α-tubulin (ab18251; Abcam, Cambridge, UK); mouse monoclonal anti-nestin (#33475; Cell Signaling, Danvers, MA, USA); rabbit polyclonal anti EB3 (AB6033; Merck-Millipore, Darmstadt, Germany); AlexaFluor 488-conjugated donkey anti-mouse (A21202); AlexaFluor 488-conjugated goat anti-rabbit (A11008); AlexaFluor568-conjugated goat anti-mouse (A11031); AlexaFluor568-conjugated donkey anti-rabbit (A10042) (Thermo Fisher Scientific, Waltham, MA, USA).

### 4.2. Cell Culture and Immunofluorescence

The human mesothelial cell line MeT-5A (purchased from ATCC/LGC Standards, Teddington, UK) and the malignant mesothelioma cell lines STAV-FCS, ZL34 and M-14-K were cultured in RPMI medium supplemented with 5% fetal bovine serum, 5% bovine serum and 1% penicillin/streptomycin. The malignant mesothelioma cell line Mero-25 (a generous gift from Dr. Paraskevi Heldin, Department of Medical Biochemistry and Microbiology, Uppsala University, Uppsala, Sweden) was cultured in Ham’s F10 medium supplemented with 15% fetal bovine serum and 1% penicillin/streptomycin. The malignant mesothelioma cell lines JL-1 and DM-3 were cultured in NTCT-109 medium supplemented with 20% fetal bovine serum, 1% L-glutamine and 1% penicillin/streptomycin. STAV-AB cells were cultured in RPMI medium supplemented with 10% human AB serum and 1% penicillin/streptomycin. BJ/SV40T cells were cultured in DMEM supplemented with 10% fetal bovine serum and 1% penicillin/streptomycin. All cell lines were cultured at 37 °C in 5% CO_2_. For immunofluorescence analysis, the cells were fixed in 3% paraformaldehyde in phosphate-buffered saline (PBS) for 25 min at 37 °C, and then washed in PBS. Next, the cells were permeabilized in 0.2% Triton X-100 in PBS for 5 min, washed in PBS, and incubated in 5% fetal bovine serum in PBS for 30 min at room temperature. The primary and secondary antibodies were diluted in PBS containing 5% fetal bovine serum. The cells were incubated with the primary and secondary antibodies for intervals of 1 h, followed by washing in PBS. The coverslips were mounted on object slides using of Fluoromount-G (Southern Biotechnology Associates, (Birmingham, AL, USA) The cells were photographed using a microscope (AxioVert 40 CFL; Zeiss, Oberkochen, Germany) equipped with a digital camera (AxioCAM MRm; Zeiss, Oberkochen, Germany) and the AxioVision software.

### 4.3. Wound Closure Assay

Cells were seeded in 96 well plates on cell densities that ensured confluent monolayers the day after seeding. Scratches were introduced using a wound maker, which creates wounds of equal width. Images were acquired every 30 min using a 10× phase-contrast objective and monitored using the IncuCyte Zoom imaging system (Essen Bioscience, Ann Arbor, MI, USA). The wound closure analysis software allowed quantification of the increasing cell confluence inside the wound.

### 4.4. Invasion Assay

The invasive properties of the cells were determined using invasion chambers (BioCoat Matrigel, (Thermo Fisher Scientific, Waltham, MA, USA)) with 8.0 μm polyethylene terephthalate membrane, used as described by the manufacturer. The inserts were thawed at room temperature and then incubated with serum-free medium for 2 h. After this incubation, the cells were seeded at 5 × 10^4^ cell/mL, and incubated for 20 h at 37 °C in 5% CO_2_. After this incubation, non-invading cells were removed from the inside of the insert by swabbing with a cotton swab. Next, the inserts were fixed in 3% paraformaldehyde for 5 min at 37 °C, washed with PBS, permeabilized with 0.2% Triton X in PBS, and washed with PBS. The cells were then incubated with DAPI for 20 min, the membranes were cut out of the insert housing, mounted on object slides, and covered with cover slips. For quantification, 10 images were acquired from random positions, and the number of nuclei were counted using ImageJ.

## 5. Conclusions

Our data suggest that it is possible to classify MM cell lines into separate categories using quite simple cell staining and analysis of the morphological and invasive capacity of MM cells. We hypothesize that this information could potentially be used in MM diagnostics and/or prognostics. Different treatment options might be effective for cells with high capacity to degrade extracellular matrix compared to cells with high invasive capacity but lack of matrix degrading capacity.

## Figures and Tables

**Figure 1 cancers-13-00685-f001:**
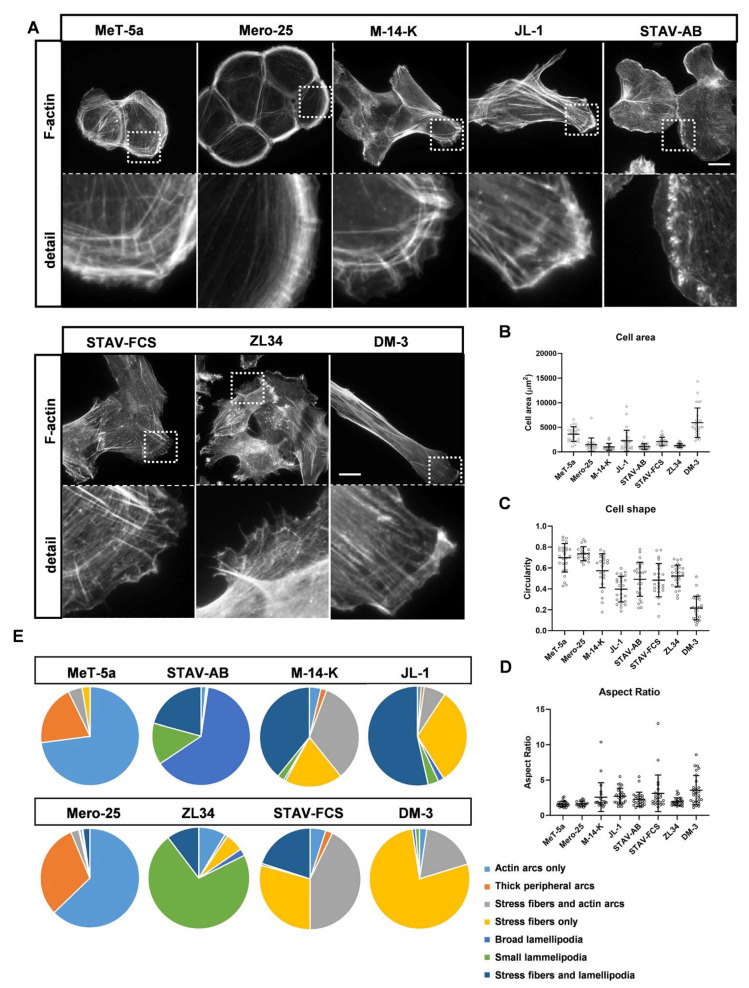
Actin filament organization and cell morphology. Representative images (**A**) for filamentous actin visualized with TRITC-conjugated phalloidin. Scale bar, 20 µm. (**B**–**D**). Quantification of cell area (**B**), circularity (**C**) and Aspect Ratio (**D**) was performed by ImageJ. MeT-5a (*n* = 25), Mero-25 (*n* = 20), M-14-K (*n* = 23), JL-1 (*n* = 26), STAV-AB (*n* = 26), STAV-FCS (*n* = 26), ZL34 (*n* = 27), DM-3 (*n* = 27). (**E**). Analysis of actin organization. 100 cells per cell lines were examined according to the dominant phenotype for each cell. The experiment was performed three times, i.e., in total 300 cells per cell lines were analyzed. One-way ANOVA with Tukey’s post hoc analyses were made to calculate statistical significance in panels (**B**–**D**). A complete list of the significances of the data is presented in Appendix A.

**Figure 2 cancers-13-00685-f002:**
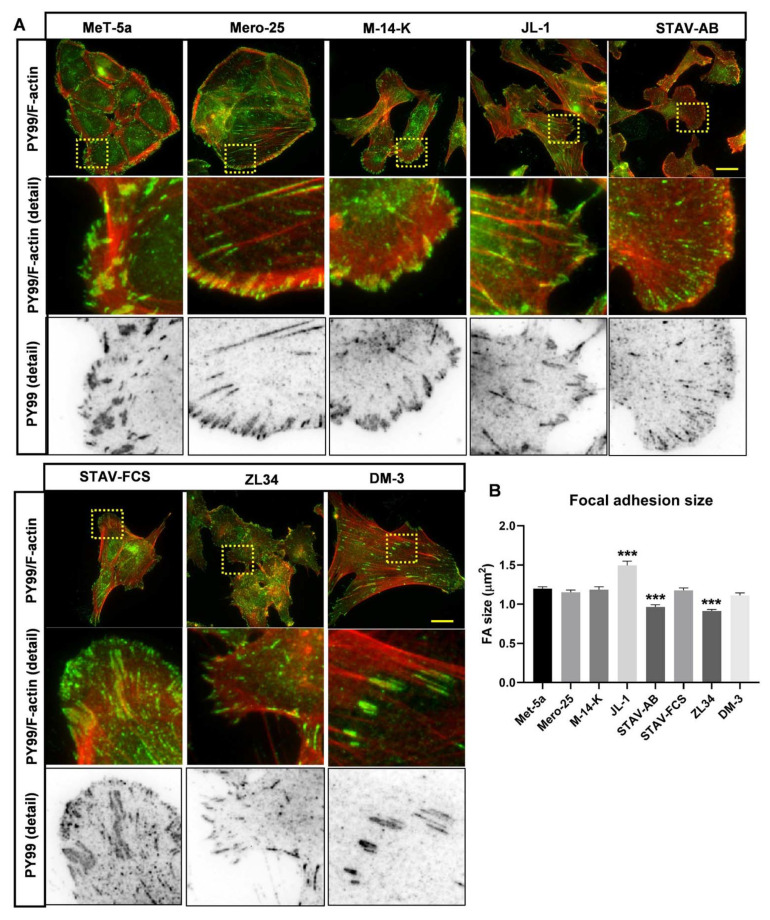
Organization of focal adhesions. (**A**). Representative images for filamentous actin visualized with TRITC -conjugated phalloidin, and for focal adhesions stained with mouse anti-phosphotyrosine (PY99) antibody followed by a AlexaFluor488-conjugated anti-mouse antibody. Scale bar, 20 µm. (**B**). Focal adhesion size were calculated using ImageJ. MeT-5a (*n* = 4415), Mero-25 (*n* = 3995), M-14-K (*n* = 2025), JL-1 (*n* = 1454), STAV-AB (*n* = 2813), STAV-FCS (*n* = 2733), ZL34 (*n* = 5637), DM-3 (*n* = 2513). Error bars denotes standard error of mean. One-way ANOVA with Tukey’s post hoc analyses were made to calculate statistical significance. *** = *p* < 0.001 marks the statistical significant difference compared to MeT-5a. A complete list of the significances of the data is presented in Appendix A.

**Figure 3 cancers-13-00685-f003:**
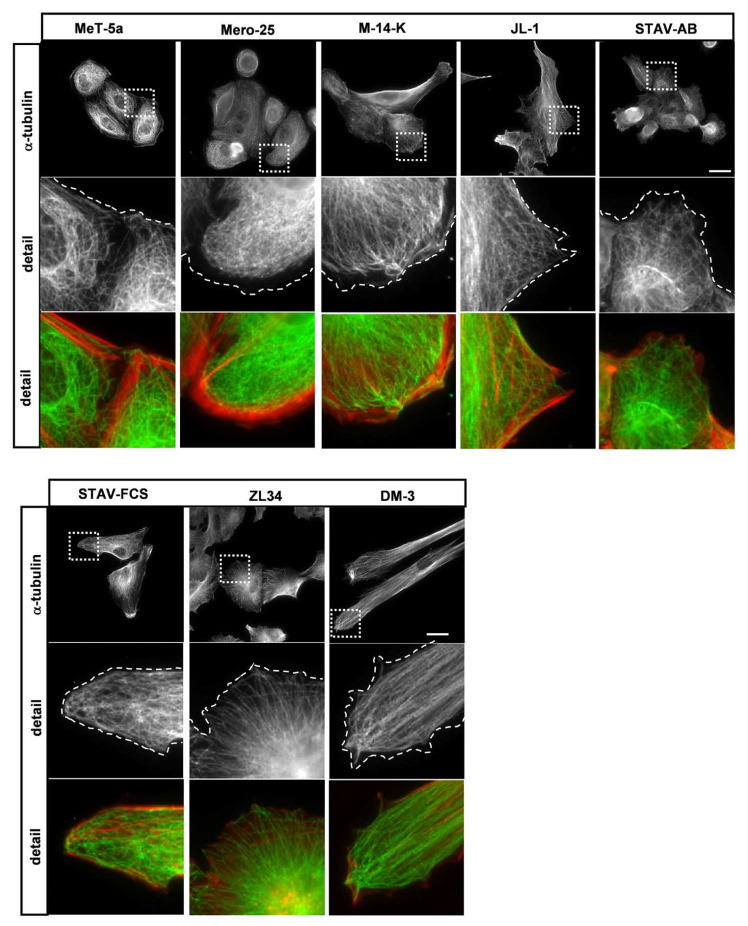
Organization of microtubules. Representative images for microtubule visualization with a mouse anti–α-tubulin antibody followed by an AlexaFluor488-conjugated goat anti-mouse antibody. Filamentous actin visualized with TRITC-conjugated phalloidin, Scale bar, 20 µm. The dashed lines mark the location of the cell edge.

**Figure 4 cancers-13-00685-f004:**
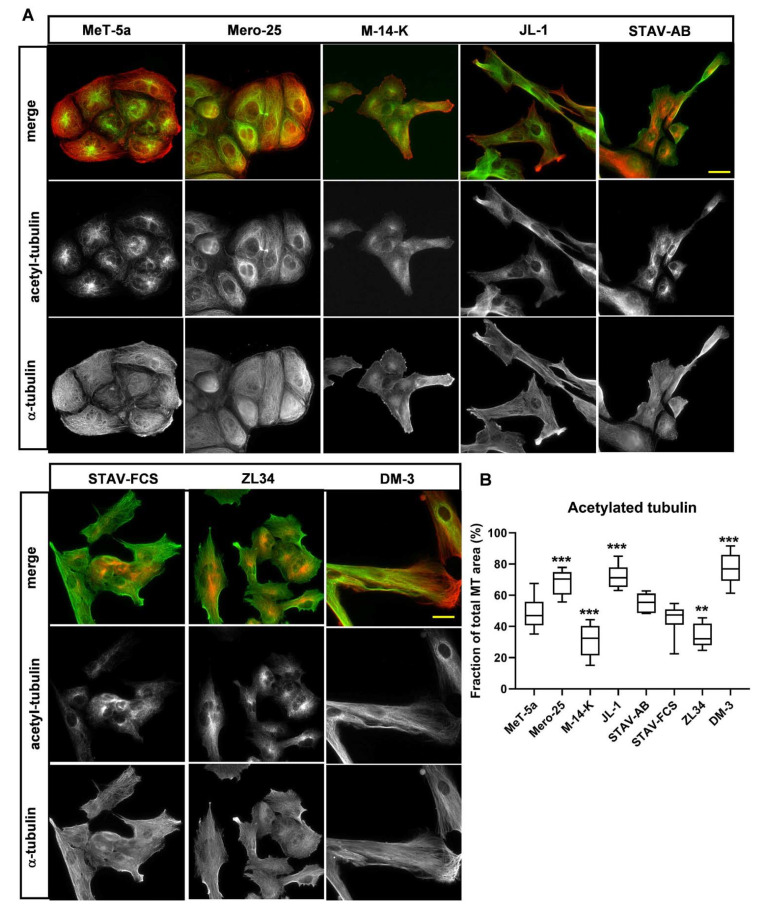
Organization of stable microtubules. (**A**) Stable microtubules were visualized with a mouse antibody against acetylated tubulin followed by an AlexaFluor488-conjugated goat anti-mouse antibody (for MeT-5a, Mero-25, M-14-K, JL-1 and DM-3) or an AlexaFluor568-conjugated goat anti-mouse antibody (for STAV-AB, STAV-FCS and ZL34). Microtubules were visualized with a rabbit antibody against acetylated tubulin followed by an AlexaFluor568-conjugated donkey anti-rabbit antibody (for MeT-5a, Mero-25, M-14-K, JL-1 and DM-3) or an AlexaFluor488-conjugated goat anti-rabbit antibody (for STAV-AB, STAV-FCS and ZL34). Scale bar, 20 µm. (**B**) The proportion of the microtubules that are also positive for acetylated tubulin was examined by microscopy imaging. Ten images from three independent experiments for each cell line were analyzed using ImageJ. One-way ANOVA with Tukey’s post hoc analyses were made to calculate statistical significance. ** = *p* < 0.01 and *** = *p* < 0.001 mark the statistical significant difference compared to MeT-5a. A complete list of the significances of the data is presented in Appendix A.

**Figure 5 cancers-13-00685-f005:**
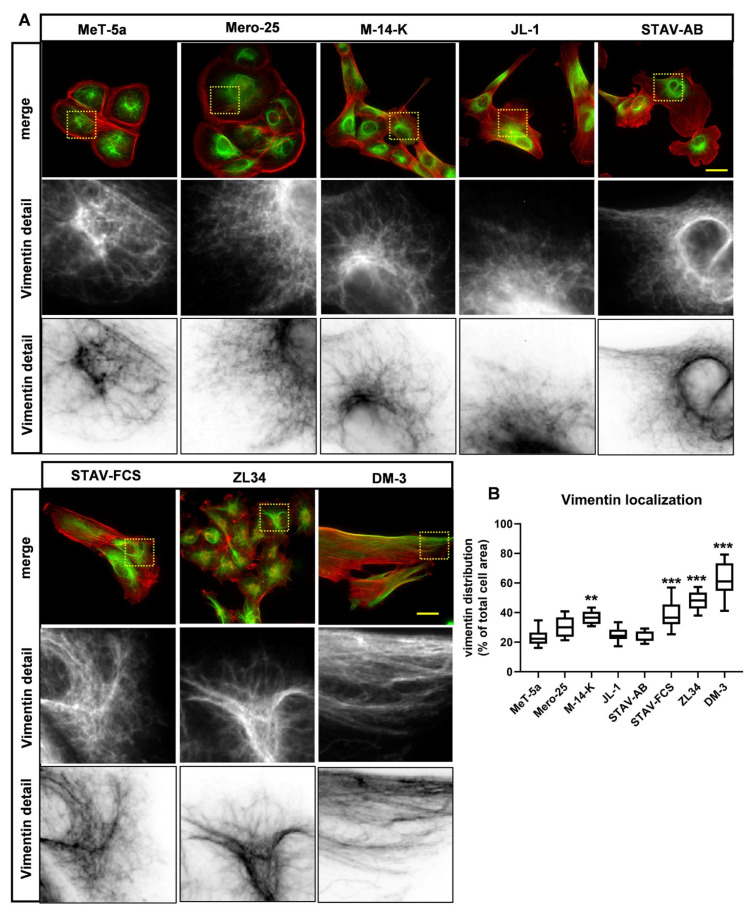
Organization of vimentin filaments. (**A**). Representative images vimentin intermediate filaments visualized with a mouse anti-vimentin antibody followed by an AlexaFluor488-conjugated anti-mouse antibody. Filamentous actin visualized with TRITC-conjugated phalloidin, Scale bar, 20 µm. (**B**). The cell area occupied by vimentin filaments was examined by microscopy imaging. Ten images from three independent experiments for each cell line were analyzed using ImageJ. One-way ANOVA with Tukey’s post hoc analyses were made to calculate statistical significance. ** = *p* < 0.01 and *** = *p* < 0.001 mark the statistical significant difference compared to MeT-5a. A complete list of the significances of the data is presented in Appendix A.

**Figure 6 cancers-13-00685-f006:**
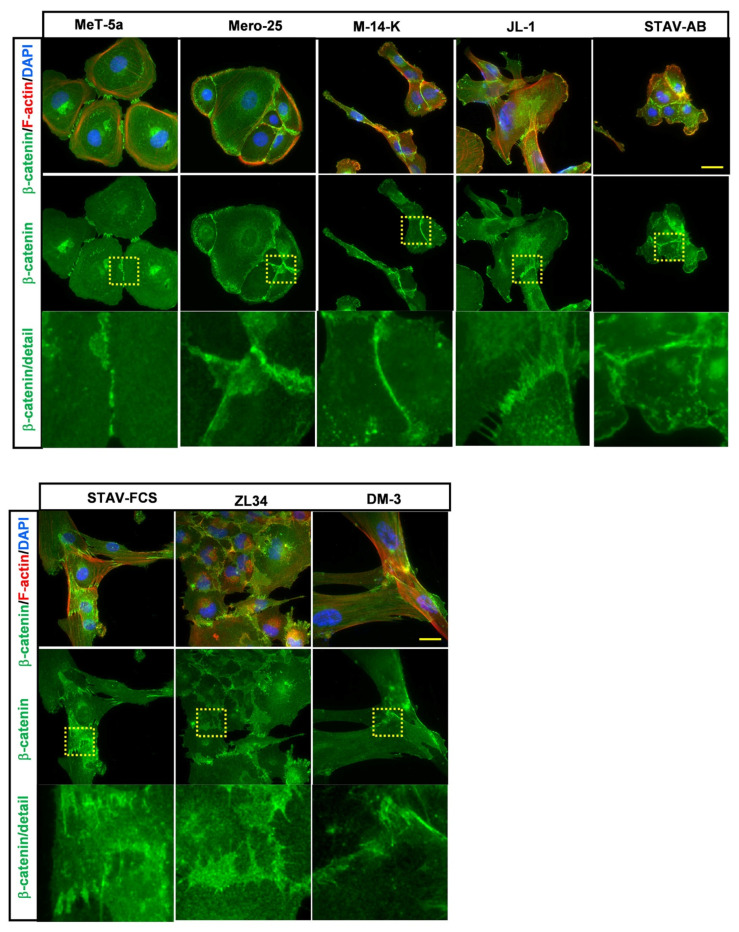
Organization of β-catenin. Representative images for the cell:cell contact areas visualized with a mouse anti–β-catenin antibody followed by an AlexaFluor488-conjugated anti-mouse antibody. Filamentous actin visualized with TRITC -conjugated phalloidin, Scale bar, 20 µm.

**Figure 7 cancers-13-00685-f007:**
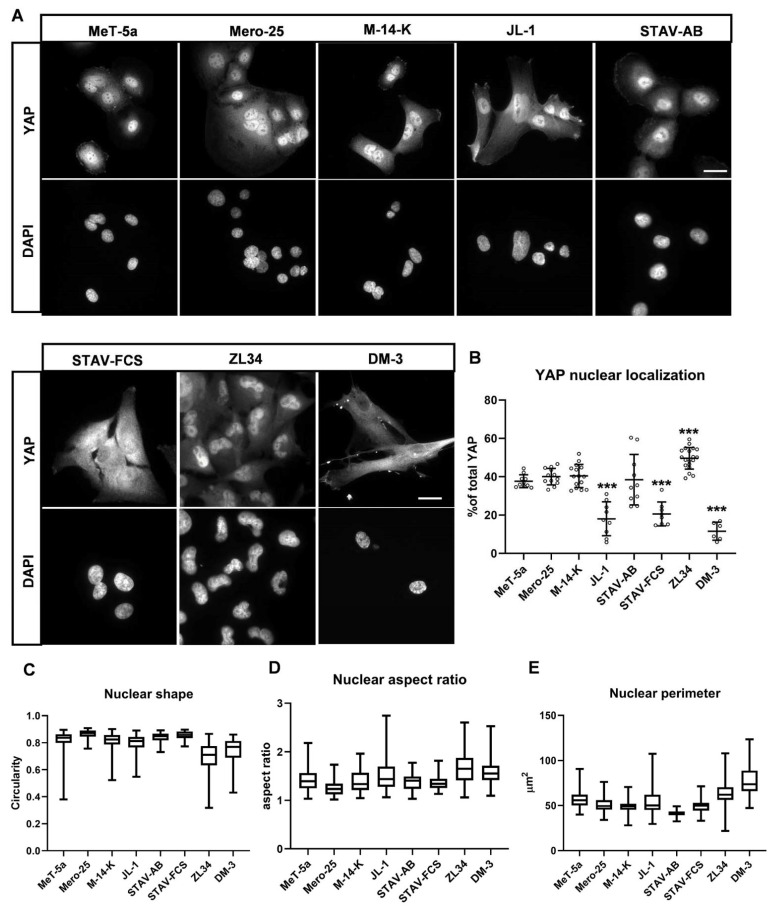
YAP nuclear localization and nuclear shape. (**A**,**B**). Representative images (**A**) and quantification (**B**) of YAP nuclear localization analyzed by immunofluorescence microscopy after visualization with a mouse anti-YAP antibody followed by an AlexaFluor488-conjugated anti-mouse antibody. Nuclei were visualized with DAPI. Scale bar, 20 µm. The Proportion of nuclear YAP nuclear YAP over the total cellular YAP was deduced from the microscopy images using ImageJ. MeT-5a (*n* = 10), Mero-25 (*n* = 12), M-14-K (*n* = 15), JL-1 (*n* = 9), STAV-AB (*n* = 8), STAV-FCS (*n* = 8), ZL34 (*n* = 19), DM-3 (*n* = 6). (**C**–**E**). Quantification of nuclear shape (circularity, **C**), aspect ratio (**D**) and perimeter (**E**) was performed by ImageJ. MeT-5a (*n* = 72), Mero-25 (*n* = 62), M-14-K (*n* = 92), JL-1 (*n* = 68), STAV-AB (*n* = 77), STAV-FCS (*n* = 50), ZL34 (*n* = 94), DM-3 (*n* = 59). One-way ANOVA with Tukey’s post hoc analyses were made to calculate statistical significances of the data in panel (**B**–**E**). *** = *p* < 0.001 mark the statistical significant difference compared to MeT-5a. A complete list of the significances of the data is presented in Appendix A.

**Figure 8 cancers-13-00685-f008:**
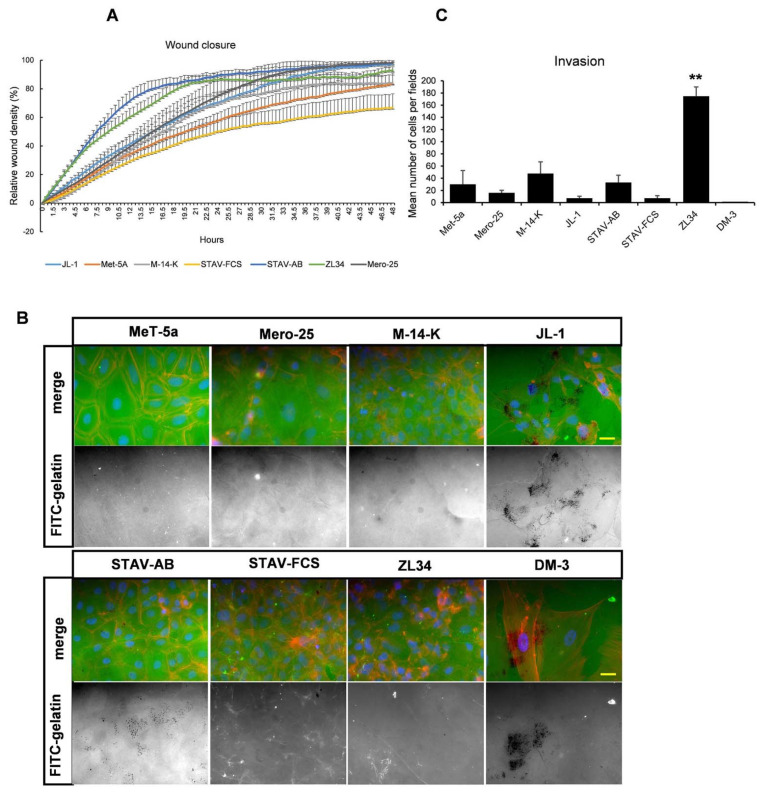
Migratory properties. (**A**). Quantification of wound closure with time as followed by live-cell imaging over 48 h using an IncuCyte imaging device. (**B**). Representative images of matrix degradation activity of MM cell lines 48 h after seeding on coverslips coated with FITC-labelled gelatin. Scale bar, 20 µm. (**C**). Quantification of the migration of STAV-FCS, STAV-AB, and ZL34 cells through Boyden-type filter chambers coated with Matrigel, in terms of invasion index. Unpaired two-way Student’s *t*-tests with unequal variance were performed to calculate statistical significance for the data. ** = *p* < 0.01 marks the statistical significant difference compared to MeT-5a.

## Data Availability

Data is contained within the article or Supplementary Material.

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
