# Peer review of "Cytoskeletal Organization Correlates to Motility and Invasiveness of Malignant Mesothelioma Cells"

_cancers, 2021, doi:10.3390/cancers13040685_

Round 1

Reviewer 1 Report

Keller and colleagues provide a comprehensive assessment of cytoskeletal features of a wide panel of mesothelioma cell lines. In particular, they investigate the organization of actin filaments (describing the formation of actin arc, stress fibers, broad/small lamellipodia); of focal adhesions; of microtubules and their acetylated status; of vimentin filaments and of cell-cell contacts through beta catenin staining. They also evaluate the subcellular localization of YAP, a downstream player of the Hippo pathways the role of which is well studied in mesothelioma. Finally, they investigate migration and invasion properties of these cell lines and discuss how the analyzed cytoskeletal features affect these tumorigenic properties.

The study has several flaws:

-it is very descriptive and lacks functional experiments to validate the correlation between phenotype and biological behavior;

- there is only one cell line frankly derived from a mesothelioma diagnosed as sarcomatoid;

- the cell behavior of established cell cultures in 2D might not recapitulate the more complex in vivo setting.

Nonetheless, the study is well written, the fluorescent images look nice and the interpretation of the immunostaining is meaningful. Overall, the study is quite comprehensive of this aspect and, as such, can be considered a useful resource of interest for those studying this deadly tumor.

I have just a few suggestions.

  1. Line 37, the authors introduce MM as a mesenchymal tumor; as far as I know this tumor is quite peculiar being derived from the mesoderm rather than endoderm. Also, one of the latest WHO classification of tumors of the Pleura distinguishes clearly MM from mesenchymal tumors: https://core.ac.uk/download/pdf/82761513.pdf. I’d recommend the authors to check carefully this issue.
  2. Consistently, various markers that usually characterize the epithelial-mesenchymal transition typical of other tumors, are differently expressed in mesothelioma. It could be interesting if the authors could introduce more thoroughly this aspect;
  3. it would be useful if the authors could comment on the ability of these cell lines to grow as spheroids or to form xenografts, if known
  4. it would be very useful if the authors could provide a table summing up all the info emerged for each cell line and eventually features retrieved from the literature.

Minor issues:

Line 40: change to abdominal MMs

Line 44: change to “there are three main phenotypic variants of MM”

Line 69 and 70: add references to support these statements

Line 74:  remove ‘excessive’  maybe change to ‘extensive’

Figure 1: panels B,C,D are quite small…is the resolution high enough?

Do the authors refer to YAP1? Are there any other variants that might be confused? Please specify if so

Line 388: the authors in a mat & methods heading refer to transfection but no such experiment is performed right? Please check

I could not find the tables reporting the statistics

Some arrows pointing to various structures could be useful to guide non expert readers in the interpretation of some images (for example actin caps, )

Author Response

Reviewer #1

Keller and colleagues provide a comprehensive assessment of cytoskeletal features of a wide panel of mesothelioma cell lines. In particular, they investigate the organization of actin filaments (describing the formation of actin arc, stress fibers, broad/small lamellipodia); of focal adhesions; of microtubules and their acetylated status; of vimentin filaments and of cell-cell contacts through beta catenin staining. They also evaluate the subcellular localization of YAP, a downstream player of the Hippo pathways the role of which is well studied in mesothelioma. Finally, they investigate migration and invasion properties of these cell lines and discuss how the analyzed cytoskeletal features affect these tumorigenic properties.

The study has several flaws:

-it is very descriptive and lacks functional experiments to validate the correlation between phenotype and biological behavior;

- there is only one cell line frankly derived from a mesothelioma diagnosed as sarcomatoid;

- the cell behavior of established cell cultures in 2D might not recapitulate the more complex in vivo setting.

Nonetheless, the study is well written, the fluorescent images look nice and the interpretation of the immunostaining is meaningful. Overall, the study is quite comprehensive of this aspect and, as such, can be considered a useful resource of interest for those studying this deadly tumor.

We thank the reviewer for the positive comments.

I have just a few suggestions.

  1. Line 37, the authors introduce MM as a mesenchymal tumor; as far as I know this tumor is quite peculiar being derived from the mesoderm rather than endoderm. Also, one of the latest WHO classification of tumors of the Pleura distinguishes clearly MM from mesenchymal tumors: https://core.ac.uk/download/pdf/82761513.pdf. I’d recommend the authors to check carefully this issue.

We have removed “mesenchymal” in order to avoid confusion

  1. Consistently, various markers that usually characterize the epithelial-mesenchymal transition typical of other tumors, are differently expressed in mesothelioma. It could be interesting if the authors could introduce more thoroughly this aspect;

We agree with the reviewer that this is a very interesting aspect. However, our study is focusing on differences in cytoskeletal organization and function. Therefore, an extensive discussion about the differences between EMT and what could be called mesothelial-mesenchymal-transition is outside the scope of this article.

  1. it would be useful if the authors could comment on the ability of these cell lines to grow as spheroids or to form xenografts, if known

STAV-AB and STAV-FCS can grow in xenografts and can also grow in spheroids (see. Front Oncol. 2013 Aug 9;3:203. doi: 10.3389/fonc.2013.00203. eCollection 2013.). It is not known to us if the other cell lines can grow as spheroids, but this is something our continued studies could show.

  1. it would be very useful if the authors could provide a table summing up all the info emerged for each cell line and eventually features retrieved from the literature.

We agree with the reviewer that it could be valuable to summarize all the features in one table, if possible. We have summarized the actin features in the pie representations in Fig.1, which we found useful to visualize complex characteristics. However, we found it difficult to condense all the data into one table or figure in an intelligible way. We have included a table in the supplementary information to show the origin and basic features of each of the cell lines.

Minor issues:

Line 40: change to abdominal MMs

We have changed the text accordingly.

Line 44: change to “there are three main phenotypic variants of MM”

We have changed the text accordingly.

Line 69 and 70: add references to support these statements

We have included a reference here.

Line 74:  remove ‘excessive’  maybe change to ‘extensive’

We have changed the text to “extensive”, as suggested.

Figure 1: panels B,C,D are quite small…is the resolution high enough?

We have increased the size of panels and we hope that this has increased the visibility.

Do the authors refer to YAP1? Are there any other variants that might be confused? Please specify if so

Yes, we mean YAP1 but for simplicity we refer to it as YAP throughout the text. We have clarified this in the revised Results, and Materials and Methods sections.

Line 388: the authors in a mat & methods heading refer to transfection but no such experiment is performed right? Please check

Yes, the reviewer is correct. We have corrected this mistake. We have also removed “constructs” in 4.1, since we have not used any DNA constructs.

I could not find the tables reporting the statistics

We are grateful for the reviewer for pointing this out! We forgot to submit the supplementary tables with the statistics. This material has been enclosed in the revised submission.

Some arrows pointing to various structures could be useful to guide non expert readers in the interpretation of some images (for example actin caps, )

We have added arrows to the panel showing actin caps.

Reviewer 2 Report

Dear Editor,

In the manuscript "Cytoskeletal organization correlates to motility and invasiveness of malignant mesothelioma cells" by Keller M et al, the authors very nicely describe differences found in the cytoskeletal organization of different mesothelial cell lines. The methodology and experiments are very elegantly and adequately done. I have actually just one bigger issue, but that I am sure could be answered and put in a discussion section:   1. How do they explain that in biphasic cell cultures there are no differences in 2 cell populations? Or are there any that they have noticed, and was that analyzed here and how exactly was that done in cell cultures?   I would also suggest rephrasing the conclusion and abstract since these results have to be confirmed in real-life samples of mesothelioma and normal, reactive changes, and only then we might discuss if that has some prognostic value. Based on the presented results it is hard to state this. The real value here is also in a much better characterization of different cell lines, which might help to explain some differences in other publications using those cell lines, and also enable better planning of future in-vitro experiments/analyses.   Two minor points- in the Introduction I would suggest using a bit more recent data about the number of diagnosed mesothelioma cases or incidence, not the one for the 2003-2008 period. And the other thing, I would suggest moving part with BAP-1 loss to the same sentence with Homozygous p16 deletion as a feature of a malignant mesothelial tumor. (In the second paragraph of Introduction)   Kind regards      

Author Response

Reviewer #2

In the manuscript "Cytoskeletal organization correlates to motility and invasiveness of malignant mesothelioma cells" by Keller M et al, the authors very nicely describe differences found in the cytoskeletal organization of different mesothelial cell lines. The methodology and experiments are very elegantly and adequately done.

We that the reviewer for the positive comments.

I have actually just one bigger issue, but that I am sure could be answered and put in a discussion section:   1. How do they explain that in biphasic cell cultures there are no differences in 2 cell populations? Or are there any that they have noticed, and was that analyzed here and how exactly was that done in cell cultures?

We found that the biphasic cell lines were quite homogenous, we rarely found cells that were both epithelioid and sarcomatoid in the same culture, rather, the cells had a features of both epithelioid and sarcomatoid characteristics. Therefore, we analyzed all cells in random fields of views and the data. We hope that this answers the question by the reviewer.

I would also suggest rephrasing the conclusion and abstract since these results have to be confirmed in real-life samples of mesothelioma and normal, reactive changes, and only then we might discuss if that has some prognostic value. Based on the presented results it is hard to state this.

We agree that we should tone this statement down, which we have done in the revised text. In the next phase of the project, delayed by the current Covid19 situation, will start to analyze material obtained directly from the pathology clinic.

The real value here is also in a much better characterization of different cell lines, which might help to explain some differences in other publications using those cell lines, and also enable better planning of future in-vitro experiments/analyses.   Two minor points- in the Introduction I would suggest using a bit more recent data about the number of diagnosed mesothelioma cases or incidence, not the one for the 2003-2008 period.

We have replaced this article with one from 2020 that discusses the world-wide situation, rather than the situation in the US.

And the other thing, I would suggest moving part with BAP-1 loss to the same sentence with Homozygous p16 deletion as a feature of a malignant mesothelial tumor. (In the second paragraph of Introduction)  

We do not think this is appropriate. The sentence describing homozygous p16 refers to malignant tumors in general and the sentence describing BAP1 refers to malignant mesothelioma in particular.